evolution/genetics

ageing, longevity, sex differences, reproduction, genetic background

**Author for correspondence:**
Jessica M. Hoffman
e-mail: jmhoffm@uab.edu

# Sex, mating and repeatability of *Drosophila melanogaster* longevity

Jessica M. Hoffman[1], Sophie K. Dudeck[2],
Heather K. Patterson[1] and Steven N. Austad[1]

[1]Department of Biology, University of Alabama at Birmingham, Birmingham, AL, USA
[2]Alabama School of Fine Arts, Birmingham, AL, USA

 JMH, 0000-0002-9487-1629

Costs of reproduction are seemingly ubiquitous across the animal kingdom, and these reproductive costs are generally defined by increased reproduction leading to decreases in other fitness components, often longevity. However, some recent reports question whether reproductive costs exist in every species or population. To provide insight on this issue, we sought to determine the extent to which genetic variation might play a role in one type of reproductive cost—survival—using *Drosophila melanogaster*. We found, surprisingly, no costs of reproduction nor sex differences in longevity across all 15 genetic backgrounds in two cohorts. We did find significant variation within some genotypes, though these were much smaller than expected. We also observed that small laboratory changes lead to significant changes in longevity within genotypes, suggesting that longevity repeatability in flies may be difficult. We finally compared our results to previously published longevities and found that reproducibility is similar to what we saw in our own laboratory, further suggesting that stochasticity is a strong component of fruit fly lifespan. Overall, our results suggest that there are still large gaps in our knowledge about the effects of sex and mating, as well as genetic background and laboratory conditions on lifespan reproducibility.

## 1. Background

Life-history theory posits that animals are able to acquire a finite number of resources; thus, individuals must allocate those resources among evolutionarily important fitness components such as growth, somatic maintenance, reproduction or survival [1,2]. A major implication of this hypothesis is that if an organism reproduces, its lifespan may be shortened, or some other fitness component compromised. Costs of reproduction have been witnessed for multiple life-history traits across

multiple taxa, yet these costs do not always manifest as directly observable decreases in longevity, especially as longevity is often difficult to measure in wild populations. Reductions in immune function [3,4], growth rate [5,6], increased oxidative damage [7,8], as well as physical damage [9] can also attend the large energetic costs of mating and producing offspring, and have all been well documented as negative consequences of mating and reproduction.

While negative effects of reproduction on longevity were often assumed to be ubiquitous, some recent research has questioned this assumption. For example, in wild-collected *Drosophila melanogaster*, mated females lived longer than virgin females in the laboratory [10], and mated *D. virilis* males lived longer than virgins [11]. Moreover, continuously reproducing insect queens live longer than their sterile workers (reviewed in [12]). Thus, the overall effect of reproduction on longevity is more complex than is often assumed.

In addition to conflicting results on the longevity cost of reproduction, environmental conditions (e.g. laboratory versus wild) can strongly influence life-history traits, especially longevity. Dietary media have strong effects on longevity across species, with lower calorie, as well as lower protein, diets tending to increase longevity and decrease reproduction [13,14]. In addition to dietary effects, other laboratory conditions can affect model organism lifespan including temperature [15], light [16] and humidity [17]. The differences between laboratories, or even within a laboratory at different times, can make comparisons of lifespan experiments difficult, as well as lead to reproducibility issues concerning the impact of potential lifespan-extending interventions [18–20].

Costs of reproduction, as well as many other phenotypic measures, are often measured in only one genetic background in the laboratory, though genetic variation has been investigated with small sample sizes across genetic backgrounds in response to mating [21]. Especially within inbred worm, fly and mouse laboratory strains, genetic background can lead to large variations in measured phenotypes. To this end, the use of the large populations of inbred strains has been increasing over the last decade, especially in fruit flies and mice. In fruit flies, the Drosophila Genetics Reference Panel (DGRP) [22], containing 200 iso-female lines were established from multi-generational, full-sib inbreeding of the descendants of inseminated flies collected from the wild. Panels of such lines can be used to determine the effects of variable genetic backgrounds on phenotypes of interest and eventually identify genes involved in those phenotypes. Other similarly designed panels exist in flies as well as laboratory mice [23,24]. Previous studies have used the DGRP to understand genetic background effects on fecundity [25] and longevity [26,27], as well as recent work on genetic background in dietary restriction and longevity [28]. However, the role of mating in longevity, sex differences in longevity, as well as the reproducibility of longevity studies both within and across laboratories have not been fully investigated.

To begin to address these issues, we performed experiments aimed at several purposes. First, we wished to discover the extent to which genetic background plays a role in the longevity cost of reproduction in both sexes. Second, we also expected to uncover genetic variation for sex differences in longevity. Third, by repeating our experiments with a second common laboratory diet, we wished to investigate how reproducibly genetic variation affected these phenotypes under subtly altered husbandry conditions.

## 2. Methods

### 2.1. *Drosophila* stocks and husbandry

Fourteen *Drosophila melanogaster* strains were procured from the Bloomington Stock Center (12 lines from the DGRP [22] and two standard inbred laboratory strains, yw and Oregon-R). In addition, duplicate populations of one strain (yw) were evaluated. One of those populations was recently obtained from the Stock Center (ywB), and the other had been obtained from the Stock Center years ago and had been maintained in Dr Nicole Riddle's closed laboratory population at UAB ever since (ywN). Therefore, a total of the 15 genotypes were used in the experiment. DGRP lines were chosen based on previously published short, medium and long lifespans [26]. Flies were maintained at 24°C at approximately 60% humidity on a 12 L : 12 D cycle. New stocks were ordered before the second iteration of the experiment to reduce genetic drift effects within our laboratory. In the winter 2017 experiment (Cohort 1), flies were kept on commercial, proprietary Jazz-Mix media (Fisher Scientific— brown sugar, corn meal, yeast agar, benzoic acid, methylparaben, propionic acid). In Summer 2019 (Cohort 2), we made our own media, composed of 8.5% sugar (5.5% dextrose and 3% sucrose), 2.5% yeast, 6% cornmeal and 1% agar with $3 \, \text{ml} \, l^{-1}$ of propionic acid added as an anti-fungal. For both

experiments, fresh media were made every two to four weeks. Approximately 5 ml of food was placed in a vial, and fresh vials were sealed and stored at 4°C until use.

## 2.2. Longevity

For both cohorts, approximately 100 flies of each of four groups were collected for each genotype: virgin males, virgin females, mated males and mated females. Virgin flies were collected under light $CO_2$ anaesthesia within 8 h of eclosion. Mated flies were allowed to mate for 48–72 h after eclosion, and then were separated into same-sex groups under light $CO_2$ anaesthesia. At the time of sex separation, approximately 20 flies were placed in a new vial, with 5 replicate vials for each group. When flies were 5–7 days old, they were randomized into the longevity experiment, and the researcher was blinded to vial identity through the duration of the experiment. Flies were transferred into new vials with fresh food three times a week, and deaths were recorded at the time of transferring. Transferring continued until all flies were dead.

## 2.3. Statistical analysis

All analyses were completed in the program R [29], and code is provided in the electronic supplementary material. First, we used Cox proportional hazard models to determine the effects of sex, mating status and their interaction across all genotypes and both cohorts and then within each genotype, for each cohort separately. We ran the analysis both with each fly as an independent unit, as is done traditionally in fly longevity analyses, and controlling for the variation of individual vials, using vial as a random effect with the *coxme* package [30]. Flies that escaped from vials before death were removed from the analysis; this constituted less than 1% of flies in either cohort. The Cox proportional hazard model allows us to look at the effects of variables of interest on survival time [31]. Thus, it looks at the overall survival curves, not just descriptive values like medians and maximums, and unlike most other commonly used survival analyses, it allows for the testing of multiple predictors at once. When appropriate, we used a Bonferroni correction for multiple inferences which set the significance threshold at $p < 0.002$. Then, we completed non-parametric Spearman rank correlation analyses to determine how median and maximum lifespans were correlated across 2 years. Similarly, we correlated longevity for mated and virgin flies in each time point. Then using paired Wilcoxon tests on mean, median or maximum values of each group, we looked at the effects of mated versus virgin and male versus female longevities within each year individually. We then looked at how changes in lifespans varied across the years with mating status using Spearman rank correlations. Finally, we compared our mean longevities to previously published data on the DGRP lines [26,28]. For these analyses, we set $p < 0.01$ for significance. For all our correlations and paired tests, we used a non-parametric analysis because not every dataset fitted the assumptions of normality. Thus, to keep things consistent and conservative, we used the non-parametric tests throughout.

# 3. Results

Our final results consisted of the longevities of the 15 genotypes measured in both sexes and both mating statuses (mated versus virgin) in two separate cohorts (electronic supplementary material, table S1 for means and medians). Using a full Cox proportional hazard model, we found a significant effect of genotype, sex, cohort and sex by mating interaction on survival (all $p < 0.00025$, electronic supplementary material, table S1). There was only a very minor effect of mating overall (hazard coefficient = −0.65, $p = 0.023$) with a very minor increase in survival in virgin flies. The sex by mating interaction (hazard coefficient = 0.16, $p = 6.79 \times 10^{-5}$) suggests that across both cohorts and genotypes virgin males have lower survival compared with mated males with a minor effect seen in the females; however, there is large genetic variation in this result, discussed below. We found a strong correlation of genotype median (Spearman rank rho = 0.749, $p = 6.06 \times 10^{-12}$) and maximum (Spearman rank rho = 0.799, $p = 1.89 \times 10^{-14}$) longevities across the two cohorts (figure 1). Overall median longevity did not differ between cohorts (paired Wilcoxon test $p = 0.91$) despite them receiving different diets; however, maximum longevity, as calculated by the longest-lived fly, was greater in Cohort 2 as compared with Cohort 1 (paired Wilcoxon test $p = 2.14 \times 10^{-7}$), as well as lower mortality seen in our combined Cox proportional hazard model (electronic supplementary material, table S1). Similarly, running a linear model on cohort, sex and mating status finds no significant effect on median lifespan ($p > 0.46$ for all).

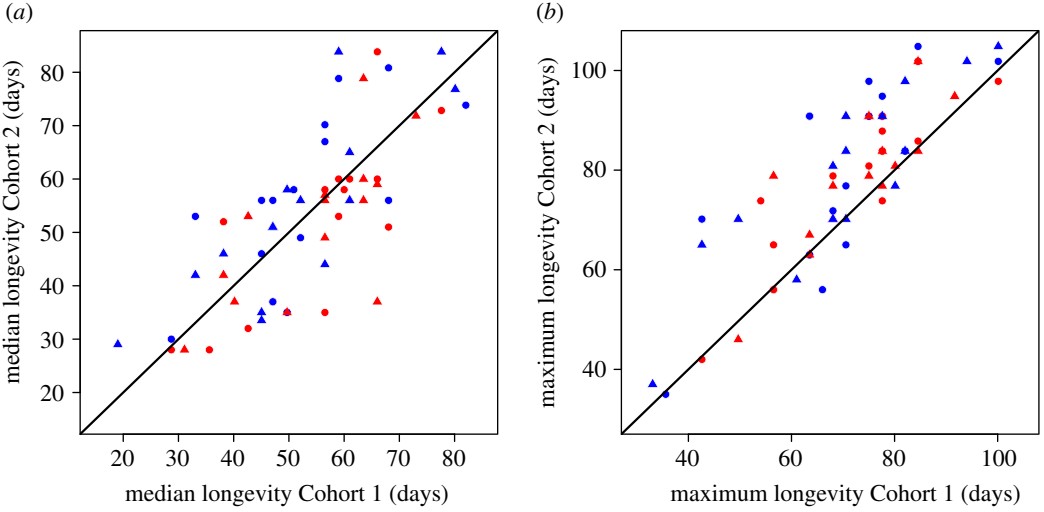

**Figure 1.** Correlation of median (*a*) and maximum (*b*) longevities between Cohorts 1 and 2. Black line is the line of symmetry. Colours indicate sex (red—female; blue—male), while points indicate virgin (triangle) and mated (circle) flies.

## 3.1. Longevity cost of reproduction

The main goal of our experiment was to evaluate genetic variation in mating costs in both sexes. To our surprise, we found no overall longevity effect of mating on either median or maximum lifespan (Wilcoxon paired test $p > 0.06$ for all four comparisons; electronic supplementary material, figure S1). While there was no *overall* impact of mating on longevity, there were significant differences *within* individual genetic lines (table 1, figures 2 and 3). For instance, in Cohort 1, only 5 of the 15 genotypes showed a significant effect of mating, in 4 of these virgins were longer-lived, in the other mated flies lived longer. In Cohort 2, 4 of the 15 genotypes displayed significant mating effects, all living longer as virgins. However, only one genetic background showed similar effects in both cohorts (RAL911). Across all genotypes, there was no correlation between the absolute differences in lifespan as a function of mating status between cohorts for either sex (figure 4). In addition, when we ran a *coxme* model (see Methods) controlling for random variation per vial, the individual effects of mating were reduced even further, with only one genotype in each cohort passing our significance threshold (electronic supplementary material, table S2). Therefore, the effects of mating on longevity appear small and sporadic with subtle environmental differences contributing more to longevity differences than reproductive status.

## 3.2. Sex differences in longevity

Previous research has consistently suggested that female *D. melanogaster* live longer than males (reviewed in [32]). However, we found that across the entire experiment, neither sex was consistently longer-lived than the other ($p = 0.12$ overall, $p = 0.25$ for Cohorts 1 and 2, respectively). Yet, similar to our mating results, we found sex differences in longevity within some individual strains, but those differences were not stable across environmental conditions (table 1, figures 2 and 3). Specifically, eights strains showed significant sex differences in Cohort 1, five in which females lived longer, three with males living longer. In Cohort 2, five strains showed significant sex differences, all favouring male longevity. Only two strains (RAL730 and RAL911) showed the same longevity differences in both cohorts. If we add in the random variation of vial, the majority of these sex differences remain significant (electronic supplementary material, table S2), suggesting that vial-to-vial variation in sex differences in lifespan within a genotype may be minimal, and less so as compared to what we saw with mating.

In addition, we found striking sex-by-mating interactions both when all cohorts are run together (electronic supplementary material, table S1) and in our individual genotype analysis (four strains in both Cohorts 1 and 2, of which only one overlapped (YWN) between cohorts). However, we should note that quite a few other genotypes were close to significant but did not pass our Bonferroni correction due to the smaller power to detect interaction effects. In general, these interactions led to decreased survival in virgin males compared with mated males, yet in other genotypes, opposite effects were seen. Therefore, genetic variation appears to play a large role in sex responses to mating. Similar to

**Table 1.** Cox proportional hazard estimates and *p*-values for each individual genotype from each cohort. Bold values indicate those estimates and *p*-values that pass our Bonferroni correction.

| cohort | genotype | sex estimate (male effect) | mating estimate (virgin effect) | interaction estimate | sex *p*-value | mating *p*-value | interaction *p*-value |
|---|---|---|---|---|---|---|---|
| 1 | ORB | **1.88125545** | 0.29429109 | −1.0203127 | **$6.69 \times 10^{-9}$** | 0.39337931 | 0.019428 |
| 1 | RAL304 | −0.2164127 | **0.42797151** | **−1.1163454** | 0.10554253 | **0.00147108** | **$2.37 \times 10^{-8}$** |
| 1 | RAL313 | **−0.7008301** | 0.20964375 | 0.10528468 | **$1.14 \times 10^{-7}$** | 0.10752776 | 0.56770048 |
| 1 | RAL315 | **1.25079921** | 0.01263183 | −0.1469954 | **$9.75 \times 10^{-18}$** | 0.92866136 | 0.44485358 |
| 1 | RAL373 | −0.426416 | −0.069187 | **1.69986283** | 0.04999977 | 0.74594852 | **$1.67 \times 10^{-7}$** |
| 1 | RAL383 | −0.8400589 | 0.0189315 | 0.54642554 | 0.0257433 | 0.9484984 | 0.1784084 |
| 1 | RAL397 | 0.20364028 | −0.3320545 | 0.08020277 | 0.11827066 | 0.00948117 | 0.66680404 |
| 1 | RAL703 | **1.26301447** | **−0.558658** | 0.54832389 | **$1.32 \times 10^{-19}$** | **$7.09 \times 10^{-5}$** | 0.00497534 |
| 1 | RAL714 | 0.13344851 | 0.26323341 | −0.2525404 | 0.37911066 | 0.06951211 | 0.21619473 |
| 1 | RAL730 | **−1.115093** | **−1.0557839** | 0.1236244 | **$4.82 \times 10^{-6}$** | **$5.18 \times 10^{-7}$** | 0.68527701 |
| 1 | RAL765 | 0.418956 | −0.0313534 | 0.93528169 | 0.09106596 | 0.90969729 | 0.01385829 |
| 1 | RAL822 | 0.32981327 | 0.27159263 | −0.422733 | 0.04380163 | 0.06228022 | 0.04642143 |
| 1 | RAL911 | **−1.1809718** | **−0.7676389** | 1.6919309 | **$4.18 \times 10^{-16}$** | **$1.62 \times 10^{-7}$** | **$5.71 \times 10^{-15}$** |
| 1 | YWB | **1.52265443** | 0.40496493 | −0.1522719 | **$2.46 \times 10^{-24}$** | 0.00292956 | 0.42239703 |
| 1 | YWN | **0.75862259** | **−0.8980879** | 1.0439282 | **$1.14 \times 10^{-8}$** | **$1.05 \times 10^{-6}$** | **$2.32 \times 10^{-5}$** |
| 2 | ORB | −0.0501577 | −0.0725343 | 0.52339377 | 0.7412284 | 0.63257918 | 0.0155878 |
| 2 | RAL304 | −0.0876073 | 0.17844015 | −0.3667462 | 0.57553225 | 0.25505443 | 0.09364461 |
| 2 | RAL313 | 0.03861692 | 0.25158461 | −0.5633572 | 0.79261625 | 0.07812984 | 0.00791138 |
| 2 | RAL315 | 0.30478014 | 0.39476461 | −0.0506702 | 0.03858638 | 0.01654323 | 0.81479773 |
| 2 | RAL373 | 0.17143656 | −0.0666056 | 0.26357076 | 0.2790193 | 0.67805537 | 0.25000055 |
| 2 | RAL383 | −0.1043499 | **−0.5532033** | **0.67173498** | 0.47635678 | **0.00022747** | **0.00172381** |
| 2 | RAL397 | **−1.4182068** | 0.336266 | −0.5208199 | **$5.96 \times 10^{-20}$** | 0.01908596 | 0.01162399 |
| 2 | RAL703 | 0.24112145 | 0.06008561 | **0.73756732** | 0.15584807 | 0.72966914 | **0.00193796** |

(*Continued.*)

**Table 1.** (*Continued*.)

| cohort | genotype | sex estimate (male effect) | mating estimate (virgin effect) | interaction estimate | sex p-value | mating p-value | interaction p-value |
|---|---|---|---|---|---|---|---|
| 2 | RAL714 | **−0.6367697** | 0.4794583 | −0.2708372 | **4.31 × 10⁻⁵** | 0.0107361 | 0.30333834 |
| 2 | RAL730 | **−2.1036203** | −0.0211873 | 0.75962133 | **8.59 × 10⁻²²** | 0.8998447 | 0.00212123 |
| 2 | RAL765 | **−0.4938272** | **−0.7811752** | **0.88541244** | **0.00066546** | **1.80 × 10⁻⁶** | **9.82 × 10⁻⁵** |
| 2 | RAL822 | 0.23649965 | −0.3179098 | 0.76422485 | 0.17514306 | 0.05284581 | 0.00248625 |
| 2 | RAL911 | **−1.4141141** | **−0.8170831** | 0.5954431 | **1.44 × 10⁻¹⁸** | **9.27 × 10⁻⁷** | 0.00949896 |
| 2 | YWB | 0.10838482 | −0.0961741 | −0.0629239 | 0.48511009 | 0.53240396 | 0.77301336 |
| 2 | YWN | −0.2102139 | −0.4556548 | **0.83684346** | 0.19472675 | 0.00519453 | **0.00028106** |

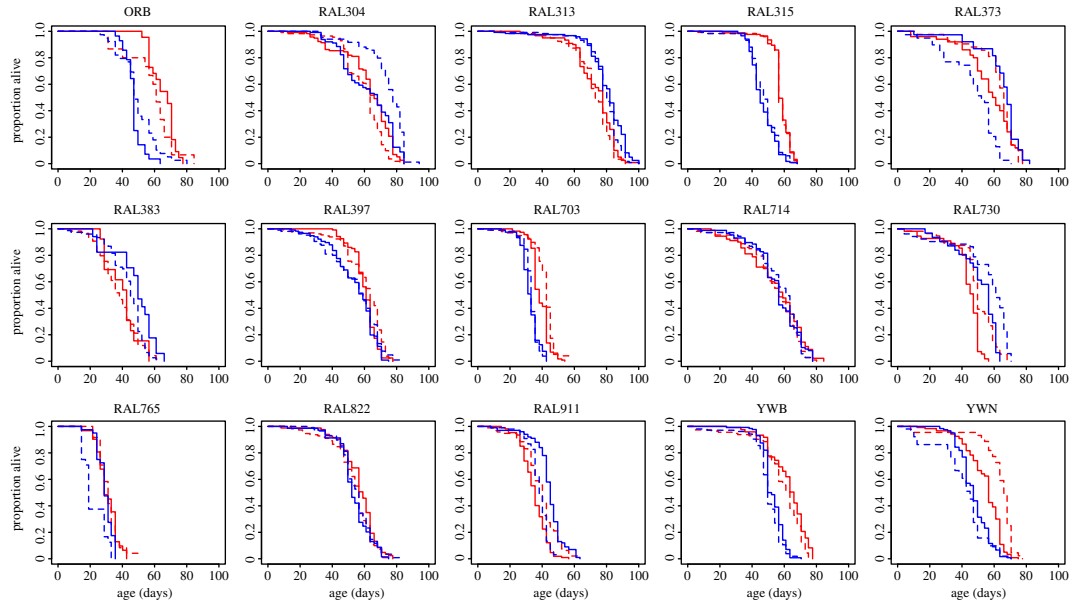

**Figure 2.** Survival curves of all Cohort 1 genotypes. Red—females and blue—males. Dashed lines are virgins. Solid lines are mated. Significant effects of sex, mating and their interaction can be seen in table 1.

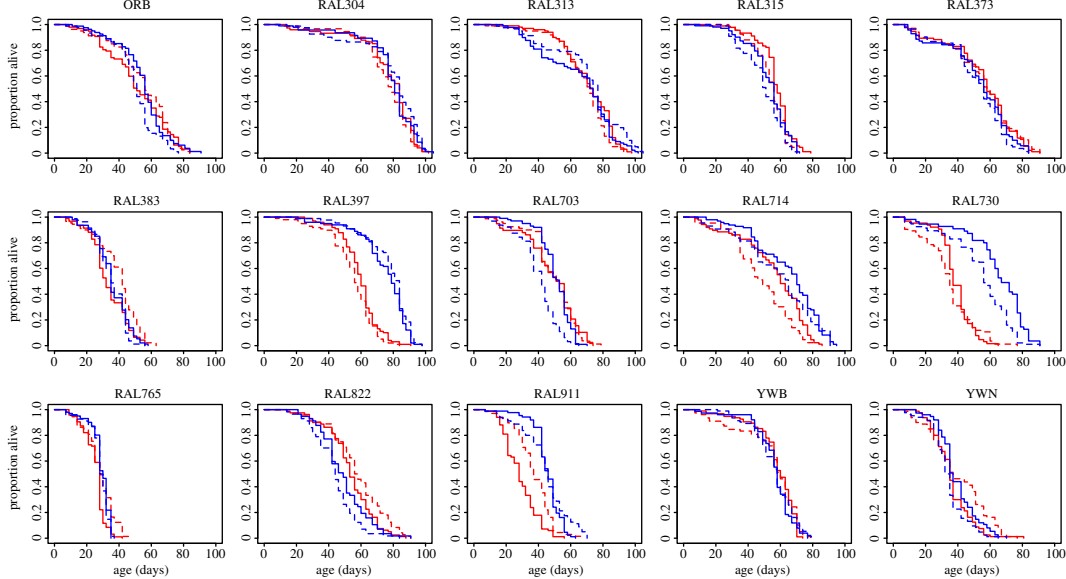

**Figure 3.** Survival curves of all Cohort 2 genotypes. Red—females and blue—males. Dashed lines are virgins. Solid lines are mated. Significant effects of sex, mating and their interaction can be seen in table 1.

both the mating and sex individual effects, the lack of reproducibility between the two cohorts in genotype effects suggests that environmental effects may matter more than interaction effects of sex and mating. Thus overall, it appears that under the conditions of our study, females do not robustly live longer. Moreover, we found that cohort effects are major factors in determining the longer-lived sex, similar to results seen in mice [32].

## 3.3. Variation in longevity across cohorts and laboratories

The differences we observed in the performance of individual lines between cohorts using subtly different diets made us curious about how our genotype longevity patterns compared with those reported in other studies under very different environmental conditions. We, therefore, compared our results with previously published longevity data for virgins of both sexes [26] and mated females [28].

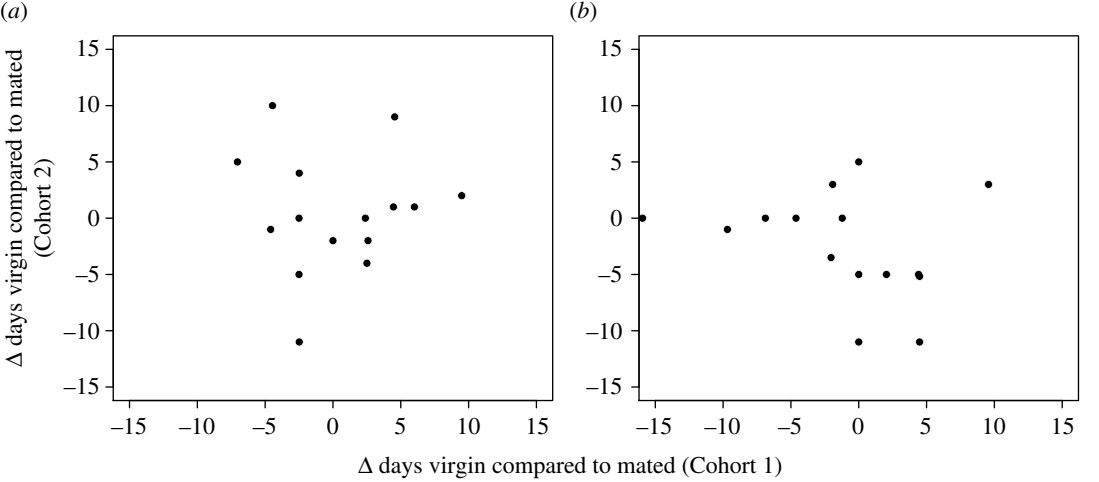

**Figure 4.** Comparison of the difference in median virgin-mated lifespan for females (*a*) and males (*b*). Each point represents one genotype. Median mated lifespan was subtracted from median virgin lifespan. No correlation exists in either sex between the years (Spearman rank $p > 0.18$ for both). Similar results are seen if mean lifespans are used (Spearman rank $p > 0.29$ for both).

**Figure 5.** Comparison of mean longevities between previously published longevities in [26] (*a*) and [28] (*b*) with the current experiment. Data in (*a*) are from virgin males (blue) and females (red), while data in (*b*) are from mated females. Black line is the line of symmetry.

We found, rather reassuringly, that longevity for both our cohorts was highly correlated with previously published results (figure 5). In the case of virgins, our mean longevities correlated nicely with results from the Mackay laboratory [26] (Cohort 1: Spearman rank rho = 0.811, $p = 3.008 \times 10^{-6}$; Cohort 2:

Spearman rank rho = 0.653, $p = 0.0007$). Similarly, for mated females, our median longevity correlated nicely with results from the Kapahi laboratory [28] (Cohort 1: Spearman rank rho = 0.884, $p = 0.0068$; Cohort 2: Spearman rank rho = 0.812, $p = 0.0037$). Surprisingly, the strength of the correlation of our Cohort 1 data was stronger with previously published data from other laboratories under dramatically different conditions than it was with our own Cohort 2 data.

## 4. Discussion

We found to our surprise that survival costs of mating were only occasionally observable. We detected it in only 9 of 30 individual survival experiments. In eight of the nine, virgins lived longer but in one mating conferred a survival advantage. These results are somewhat at variance with previous work which suggests that mating should consistently decrease survival. We also found no overall sex difference in longevity. When there were differences observed, it was at least as common for males to live longer (eight individual experiments) as females to live longer (five individual experiments). Furthermore, we found that neither mating costs when they were observed, nor sex differences, were consistent between dietary regimes. Only 1 of the 15 genotypes showed consistent mating costs across our two cohorts, and only two showed consistent sex differences in longevity across cohorts. These latter results suggest that dietary and/or seasonal husbandry effects on longevity, even within the same laboratory, can substantially affect the study of mating costs and sex differences. This latter result, the impact of subtle husbandry differences impacting sex differences in longevity, has also been seen in medflies [33,34] and laboratory mice [35]. Reassuringly, we found that the strain-specific longevity of our flies, although consistently shorter than in one previous study and consistently longer than in another, strongly correlated with the results of these other studies (figure 5).

Perhaps our most surprising result was the general lack of a mating cost on survival. We found a slight trend for longer-lived virgin flies, as would be expected and as had been shown in a previous small study in isogeneic flies [21]. However, overall, our mating effects were mild to nonexistent. The majority of these inbred fly strains showed no longevity impact from mating, and across all genotypes there was no difference between mated and virgin longevity. In addition, for those strains that did show an effect of mating, results were not consistent from one cohort to the next, with the exception of one strain. That same strain (RAL 911) was one of only two strains that showed a consistent male survival advantage across cohorts, thus it may warrant some special investigation of its resistance to environmental effects. Overall, though, small environmental effects may impact the longevity consequences of mating more than intrinsic genetic differences.

An important caveat, however, is that our study only looked at the effects of short-term mating on longevity, as flies were only allowed to mate for 48–72 h. This mating scheme allows all flies to be mated at least once, with most multiply mated, but does not cause the constant physical encounters that could lead to more physical damage of the animals, especially the females. This standard method of mating for longevity allows us to look at the intrinsic physiological effects of mating versus lifelong extrinsic damage. However, it has been suggested that female flies no longer have any sperm stored approximately two weeks after mating, though there is variation across studies (reviewed in [36]); thus, we are catching the early effects of mating, not necessarily lifelong intrinsic effects. Interestingly, work from flies captured in the wild suggests that only about 50% of females are multiply mated [37]; however, these flies were captured at a random point and not followed throughout life. Other studies have suggested that females mate on average every 27 h in the wild, and the amount of male sex peptide in the female reproductive tract declines relatively quickly [38]. Thus, while our method of mating may be comparable to the wild, the separation of males and females after 72 h will not capture the constant lifelong negative effects of mating. If we had allowed the males and females to reside together for the length of the experiment, we would expect significantly different results, and a much larger negative effect of mating, as multiple mating leads to decreased lifespan, especially in females [39]. In addition, flies were only mated to flies from the same genetic background; however, if we used a tester line to mate each line to, we potentially could have found a different response or stronger response to mating. We must note that we only used longevity as our measurement of 'costs of reproduction', and numerous other reproductive costs to fitness have been identified [40].

We also failed to find a strong sex bias in longevity. It has been generally assumed that female flies are longer lived than male flies [32], yet if anything we find a trend for increased lifespan in males in Cohort 2, though the effects are relatively minor. This is similar to another recent study of the DGRP fly lines that also found males lived longer [41], though this study used flies that were allowed

to mate throughout the experiment, thus exposing the females to constant male encounters and potential damage. Similar to our reproduction analysis, the reproducibility of sex differences between Cohort 1 and Cohort 2 was almost nonexistent, with the exception of two genotypes. This again suggests that small environmental effects may be playing a stronger role in determining sex differences than intrinsic genetic effects. We did find significant sex-by-mating interactions both in our combined model and within genotypes suggesting that sex effects differ based on whether or not a fly is mated. However, even this effect, which might have been expected, was observed in only 4 of 15 strains in each cohort and only one case of overlap between cohorts. In the entire experiment, male virgins had decreased survival in the interaction effects, and for those interactions that were significant within genotypes, this same effect direction was more common. However, there still were significant interactions in opposite directions, suggesting a potential large genetic effect on the interaction between sex and mating. Future studies with larger numbers of genetic backgrounds may be able to tease apart the individual genes that contribute to mating and sex longevity interactions.

We also ran our analysis in two ways, once with a standard method of using each fly as an independent replicate, as is traditionally done in the ageing literature in worms, flies and mice [28,42,43], and second with a Cox proportional hazard model controlling for the random variation that could occur within replicate vials of the same group. As would be expected given the reduced degrees of freedom, including the random effect of vial leads to fewer significant results, as some of the overall variation in longevity is taken up by the variation between vials. This variation can occur for both random stochastic reasons, as well as potentially microenvironmental differences within the incubators, which is why randomizing the vials across the experiment is important. As we noted in the results, the vial-to-vial variation appears to matter more for any mating effects than sex effects within genotypes, suggesting that potentially mating differences in longevity are more often controlled by stochastic processes within a genetic background than sex differences. As many ageing studies fail to control for the effects of variation in replicate vials, plates (worms) or cages (mice), many longevity results may be overambitious in their significance, and this stochastic variation might indicate why some studies are not able to be robustly reproduced in other laboratories. We must say though, that all this notwithstanding, the correlation between our longevity values and those of other laboratories was quite strong, suggesting that there is a decent inherent genetic background effect on longevity in flies; it is just the small effects that may be hard to tease apart without enormous sample sizes and replicates.

While our original goal was to understand how genetic variation played a role in costs of reproduction, we discovered strong cohort effects from one study to the next, especially across years. While we did not interrogate environmental and husbandry effects during the course of this experiment, we can surmise that these two factors were the major drivers of the differences we saw between the two replicate experiments, as genetic backgrounds were mostly comprised of iso-female lines. Recent work in antler flies has also shown that reproducibility of mortality can be difficult within the same laboratory, though this study was using a dietary supplementation regime, not mating [44].

The largest discrepancies between 2 years were seen with regards to maximum lifespans. While median lifespans were not changed to a large degree between years/seasons, maximum lifespans were significantly longer in summer 2019 in both sexes. We have several hypotheses, all related to fly husbandry, that could potentially explain this discrepancy. Our most likely, and anecdotally supported, hypothesis is that flies living in the summer are able to maintain better water homeostasis than those in the winter. Even though the incubators were set to approximately 60% humidity, we know that these often fluctuate. We reside in the Southeastern United States where environmental humidity in the winter is often under 40% but exceeds 70% in the summer. As flies age, they lose the ability to maintain their water homeostasis [45], leading to increased risk of desiccation. Therefore, maintenance at a higher humidity could cause improved water balance in old age flies, leading to potential increased lifespan. In addition, previous work in other species of arthropods suggests that longevity is highest when humidity is highest [17,46]. A second hypothesis for our observations is that the two experiments were done on slightly different media. As fly diet can have a huge impact on health and longevity [47], this could be contributing to our observed differences. However, we would have expected to see consistent shifts in both median and maximum, not just maximum, lifespan if the summer 2019 diet was significantly beneficial for fly lifespan.

Interestingly, we found that although our fly strain longevities were significantly correlated between our two cohorts, our Cohort 1 data are more closely correlated with previously published data in the same strains than with our own Cohort 2. Similar results have been witnessed, however, in mice [48] and worms [49] where reproducibility across laboratories has been found to be devilishly difficult, even controlling for minute details.

This lack of reproducibility in significant results between our two cohorts suggests that for certain questions the use of iso-female strains for determining genes that affect different phenotypes will require exquisite attention to husbandry details. The DGRP have been used over the past decade to measure dozens of different biological phenotypes with conclusions about the genes playing a causal role in the phenotypes in question. However, if small environmental perturbations can make such differences in something a fundamental as sex differences in longevity, it is possible that many phenotypes may be more sensitive to subtle environmental variation than is generally supposed.

Our results point to a large effect of genetic background on longevity as expected but a much smaller effect of mating or sex on longevity within a strain. Future studies are needed to determine which individual genes might buffer the negative physiological responses of mating, as well as which genes may be involved in sensitivity to the impact of the environment (e.g. diet, humidity) on longevity. In addition, we found that reproducibility across experiments can be difficult, even within the same laboratory, suggesting that all *Drosophila* longevity experiments need to be analysed with season and other environmental factors in mind. In addition, as the fruit fly is used as the primary model organism to test novel compounds for their lifespan-extending effect, our results suggest that reproducibility between and even within laboratories might prove difficult. Overall, we believe that our results will lay the groundwork for beginning to understand the genetic factors that influence costs of reproduction in fruit flies, with potential conservation across other species.

Data accessibility. Raw data from both cohorts uploaded as electronic supplementary material, as well as the R code used in the analysis. The previously published data pulled from other papers are available in the supplements of those papers.

The data are provided in electronic supplementary material [50].

Authors' contributions. J.M.H., S.D. and S.N.A. designed the experiment. J.M.H., S.D. and H.K.P. conducted the experiment and analysed the results. J.M.H. and S.N.A. wrote the first draft of the manuscript. All authors commented on and approved the final manuscript.

Competing interests. We declare we have no competing interests.

Funding. This work was partially funded by K99AG059920 to J.M.H. and by R01AG057434 to S.N.A.

Acknowledgements. We thank the Austad laboratory members who helped with fly maintenance and transferring during the course of this experiment, as well as Dr Nicole Riddle at UAB for allowing us the use of equipment and space. We also thank the two anonymous reviewers for their insightful comments that improved the manuscript.

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
