## [Peer Review File · Royal Society Open Science]

Review History

RSOS-210273.R0 (Original submission)

Review form: Reviewer 1

Is the manuscript scientifically sound in its present form?

Yes

Are the interpretations and conclusions justified by the results?

Yes

Is the language acceptable?

Yes

Do you have any ethical concerns with this paper?

No

Have you any concerns about statistical analyses in this paper?

Yes

Recommendation?

Accept with minor revision (please list in comments)

Comments to the Author(s)

This manuscript would be improved by a more detailed explanation of the statistical methods written in a way a non-specialist can grasp what is being tested and how. This paper is likely to be read by many researchers using *Drosophila* as a model organism that do not routinely use these tests. Citations to these statistical methods should be included.

Also, I could not find any references to a "paired Wilcox test" when I was searching pubmed and the general internet - only to a paired Wilcoxon test. Was "Wilcox" test the correct name?

Figures 1, 2, and 4 should include units (days?) for longevity on the axes.

Figure 3 should also include units on the axes.

For all figures - as there appears to be space, the spacing for each unit on the X and Y axis should be the same, producing a square rather than a rectangle, providing a clearer (more intuitive) comparison between conditions.

Figures S1 and S2 should be included as full figures to support the text.

Review form: Reviewer 2

Is the manuscript scientifically sound in its present form?

No

Are the interpretations and conclusions justified by the results?

Yes

Is the language acceptable?

Yes

Do you have any ethical concerns with this paper?

No

Have you any concerns about statistical analyses in this paper?

Yes

Recommendation?

Major revision is needed (please make suggestions in comments)

Comments to the Author(s)

This is a very interesting manuscript, in which the authors analyse patterns of survival and longevity for virgin and mated flies, of each sex, across flies (*Drosophila melanogaster*) of 15 different genotypes. They do this across 2 cohorts (separated by 2 years in time, in which each cohort was measured in a different season and using a different diet substrate). The authors show that the genotypic effect on lifespan is strong across the two cohorts (strong correlation across genotypes, over Cohorts 1 and 2, and these measures are correlated to previous measures of genotype-specific longevity measured in other labs previously (by other researchers). What was

surprising, however, is that there was no general effect of mating on longevity (the authors had predicted to observe a cost of reproduction, as gauged by reduced longevity of flies that had previously mated relative to virgin flies), and no general sex differences on longevity (the authors had predicted to observe a female-bias, typical of other studies, but if anything noted that a slight tendency of a male bias). The authors conclude that the effects of environmental variation (cohort-effects) on longevity render the study of sex differences and mating effects unreliable (subject to low levels of repeatability across time and across labs), hence promoting a cautionary tale for researchers examining the factors that moderate patterns of aging in model species. I read the paper with interest, but do have some comments that require careful attention.

Firstly, I thought that the statistical analyses may have failed to account for the hierarchical structure of the data (which would result in pseudoreplication in the analysis). Also, I am not sure why the authors can't model all factors (genotype, cohort, sex, mating status) in the one-and-the-same models (a proportional hazards model for survival, and then linear model for longevity). They should add random effects of vial identity (since ~ 20 individuals shared the same vial throughout the experiment, and these vial-sharing effects are always large and need accounting for, otherwise will in themselves lead to spurious results and interpretations of the results, since the models will treat each individual as an independent observation when they are not independent - i.e. this would lead to pseudoreplication) and ideally the mating group identity (flies were mated in groups of 100 prior to being allocated to a sex-specific vial). These analyses can be conducted in the packages `coxme` and `lme4` respectively.

For correlations, it is unclear why the authors used the nonparametric Spearman Rank correlations and paired Wilcoxon tests rather than the parametric alternatives; particularly given that in the figures, the relationships look basically linear. Why was the alpha criterion set to 0.01 for Spearman rank tests - is this based on the number of correlations you tested (Bonferroni adjusted)?

How did the authors handle escapees - flies that were lost to the experiment before they died (censored)?

I think it is interesting that there was no clearly detectable cost of reproduction on longevity. Bearing in mind that the period of sexual cohabitation was relatively short (24 hours) and early in life (Day 2-3 of adult age, when median longevity is of the order of 40 to 60 days; what is missing here is a discussion of a) how long it takes females to become sperm limited (and discussion of whether this means reproductive costs are limited to a relatively short period early in life in which females are able to lay fertilizable eggs) and b) a discussion of the natural mating rate in fruit flies in the wild.

Discussion - in the first paragraph of the discussion, the authors talk about stability of effects across dietary regimes. This statement should be modified because there were several differences between the 2 cohorts (which the authors point out later in the discussion), so it is not possible to determine whether the cohort effects are primarily mediated by dietary variation or something else.

The sentence at the top of page 11 requires rephrasing (particularly the phrase: "results failed to stay significant").

Decision letter (RSOS-210273.R0)

Dear Dr Hoffman

The Editors assigned to your paper RSOS-210273 "Sex, mating, and repeatability of *Drosophila melanogaster* longevity" have now received comments from reviewers and would like you to revise the paper in accordance with the reviewer comments and any comments from the Editors. Please note this decision does not guarantee eventual acceptance.

Please submit your revised manuscript and required files (see below) no later than 21 days from today's (ie 12-Apr-2021) date. Note: the ScholarOne system will 'lock' if submission of the revision is attempted 21 or more days after the deadline. If you do not think you will be able to meet this deadline please contact the editorial office immediately.

on behalf of Kevin Padian (Subject Editor)
openscience@royalsociety.org

Associate Editor Comments to Author:

Comments to the Author:

A number of queries and comments have been raised and made that the authors will need to address in their revision. Please closely address the referees' reports in your revision.

Reviewer comments to Author:

Reviewer: 1

Comments to the Author(s)

This manuscript would be improved by a more detailed explanation of the statistical methods written in a way a non-specialist can grasp what is being tested and how. This paper is likely to be read by many researchers using *Drosophila* as a model organism that do not routinely use these tests. Citations to these statistical methods should be included.

Also, I could not find any references to a "paired Wilcoxon test" when I was searching pubmed and the general internet - only to a paired Wilcoxon test. Was "Wilcoxon" test the correct name?

Figures 1, 2, and 4 should include units (days?) for longevity on the axes.

Figure 3 should also include units on the axes.

For all figures - as there appears to be space, the spacing for each unit on the X and Y axis should be the same, producing a square rather than a rectangle, providing a clearer (more intuitive) comparison between conditions.

Figures S1 and S2 should be included as full figures to support the text.

Reviewer: 2

Comments to the Author(s)

This is a very interesting manuscript, in which the authors analyse patterns of survival and longevity for virgin and mated flies, of each sex, across flies (*Drosophila melanogaster*) of 15 different genotypes. They do this across 2 cohorts (separated by 2 years in time, in which each cohort was measured in a different season and using a different diet substrate). The authors show that the genotypic effect on lifespan is strong across the two cohorts (strong correlation across genotypes, over Cohorts 1 and 2, and these measures are correlated to previous measures of genotype-specific longevity measured in other labs previously (by other researchers). What was surprising, however, is that there was no general effect of mating on longevity (the authors had predicted to observe a cost of reproduction, as gauged by reduced longevity of flies that had previously mated relative to virgin flies), and no general sex differences on longevity (the authors had predicted to observe a female-bias, typical of other studies, but if anything noted that a slight tendency of a male bias). The authors conclude that the effects of environmental variation (cohort-effects) on longevity render the study of sex differences and mating effects unreliable (subject to low levels of repeatability across time and across labs), hence promoting a cautionary tale for researchers examining the factors that moderate patterns of aging in model species.

I read the paper with interest, but do have some comments that require careful attention.

Firstly, I thought that the statistical analyses may have failed to account for the hierarchical structure of the data (which would result in pseudoreplication in the analysis). Also, I am not sure why the authors can't model all factors (genotype, cohort, sex, mating status) in the one-and-the-same models (a proportional hazards model for survival, and then linear model for longevity). They should add random effects of vial identity (since ~ 20 individuals shared the same vial throughout the experiment, and these vial-sharing effects are always large and need accounting for, otherwise will in themselves lead to spurious results and interpretations of the results, since the models will treat each individual as an independent observation when they are not independent - i.e. this would lead to pseudoreplication) and ideally the mating group identity (flies were mated in groups of 100 prior to being allocated to a sex-specific vial). These analyses can be conducted in the packages *coxme* and *lme4* respectively.

For correlations, it is unclear why the authors used the nonparametric Spearman Rank correlations and paired Wilcoxon tests rather than the parametric alternatives; particularly given that in the figures, the relationships look basically linear. Why was the alpha criterion set to 0.01 for Spearman rank tests – is this based on the number of correlations you tested (Bonferroni adjusted)?

How did the authors handle escapees – flies that were lost to the experiment before they died (censored)?

I think it is interesting that there was no clearly detectable cost of reproduction on longevity. Bearing in mind that the period of sexual cohabitation was relatively short (24 hours) and early in life (Day 2-3 of adult age, when median longevity is of the order of 40 to 60 days; what is missing here is a discussion of a) how long it takes females to become sperm limited (and discussion of whether this means reproductive costs are limited to a relatively short period early in life in which females are able to lay fertilizable eggs) and b) a discussion of the natural mating rate in fruit flies in the wild.

Discussion – in the first paragraph of the discussion, the authors talk about stability of effects across dietary regimes. This statement should be modified because there were several differences between the 2 cohorts (which the authors point out later in the discussion), so it is not possible to determine whether the cohort effects are primarily mediated by dietary variation or something else.

The sentence at the top of page 11 requires rephrasing (particularly the phrase: “results failed to stay significant”).

===PREPARING YOUR MANUSCRIPT===

If you have been asked to revise the written English in your submission as a condition of publication, you must do so, and you are expected to provide evidence that you have received language editing support. The journal would prefer that you use a professional language editing service and provide a certificate of editing, but a signed letter from a colleague who is a native

speaker of English is acceptable. Note the journal has arranged a number of discounts for authors using professional language editing services (<https://royalsociety.org/journals/authors/benefits/language-editing/>).

===PREPARING YOUR REVISION IN SCHOLARONE===

<https://royalsociety.org/journals/authors/author-guidelines/#supplementary-material> to include a suitable title and informative caption. An example of appropriate titling and captioning may be found at https://figshare.com/articles/Table_S2_from_Is_there_a_trade-

off_between_peak_performance_and_performance_breadth_across_temperatures_for_aerobic_sc
ope_in_teleost_fishes_/3843624.

Author's Response to Decision Letter for (RSOS-210273.R0)

See Appendix A.

RSOS-210273.R1 (Revision)

Review form: Reviewer 1

Is the manuscript scientifically sound in its present form?

Yes

Are the interpretations and conclusions justified by the results?

Yes

Is the language acceptable?

Yes

Do you have any ethical concerns with this paper?

No

Have you any concerns about statistical analyses in this paper?

No

Recommendation?

Accept as is

Comments to the Author(s)

The revisions made have addressed my previous concerns.

Review form: Reviewer 2

Is the manuscript scientifically sound in its present form?

No

Are the interpretations and conclusions justified by the results?

Yes

Is the language acceptable?

Yes

Do you have any ethical concerns with this paper?

No

Have you any concerns about statistical analyses in this paper?

Yes

Recommendation?

Major revision is needed (please make suggestions in comments)

Comments to the Author(s)

I thank the authors for the responses and revisions, and provide some further comments below.

I think the analyses of genotype median and maximum are appropriate, since they are based on genotype-means (but this should be clarified in the methods).

It's unclear as to whether the authors use the individual fly or the genotype means as the unit of statistical analysis in some of their analyses (E.g. analyses on sex differences in longevity).

The new results of the proportional hazards model (cox model) indicate a very strong mating-by-sex interaction on survival; yet this was overlooked and not scrutinised further and not mentioned again in the Results sections on Longevity Cost of Reproduction (page 32, line 40).

But, I have some comments on this Cox model and the general linear models, since these do seem to assume that every individual fly sampled was an independent data point, for which the reader will require clarification. I explain more below.

The authors argue against incorporating variance components that are currently unaccounted for in their cox and general linear models (such as Vial ID); arguing they could do this but that it would limit comparisons to other studies in the aging research community who never take account of such variance, and use similar statistical methods to those employed in this paper. I empathize with the authors on this point. But, models that account for such sources of variance are well suited to these analyses, and at least routinely used by researchers who study the evolutionary basis of ageing; and that vial-sharing effects are not just potential sources of variance, but tend to always account for a large part of the variance. Including them is required to make sure the data are not modelled at the incorrect level, resulting in pseudoreplication and anticonservative parameter estimates. The authors argument for why they don't take account of these random effects is interesting given a large premise of the current paper is to point out that previously assumed effects of cost of mating on longevity may have likely been driven small stochastic environmental changes leading to non-reproducibility of results across studies. Using an approach that would accurately model and account for unaccounted for sources of variance would help very much in course correcting the field.

Notwithstanding, prior to these powerful approaches to multi-level modelling becoming available, researchers used to calculate vial means to conduct their analysis on these means rather than on data from individual flies (thus the unit of replication was vial rather than the fly, and the analyses not pseudoreplicated). This can't be done with survival analyses (they really require random effects), but could be done with linear models; and the analysis of genotype means does seem to be the approach used by the authors in the correlation analyses (but this should be made very transparent to the reader).

In sum, at the very least, the authors should be completely transparent and specify their units of replication for their analyses in their methods for all of their analyses – i.e. are they assuming all flies within each genotype were assayed as part of the same population (i.e. vial effects and other levels of the data in which flies within a genotype that by definition share inter-dependencies are assumed not to have an effect on survival), and should also be clear about the level of replication per genotype assumed in their analyses (seems that it is the level of the fly -- individuals per genotype in the cox models and linear models).

Decision letter (RSOS-210273.R1)

Dear Dr Hoffman

The Editors assigned to your paper RSOS-210273.R1 "Sex, mating, and repeatability of *Drosophila melanogaster* longevity" have now received comments from reviewers and would like you to revise the paper in accordance with the reviewer comments and any comments from the Editors. Please note this decision does not guarantee eventual acceptance.

Please submit your revised manuscript and required files (see below) no later than 21 days from today's (ie 09-Jun-2021) date. Note: the ScholarOne system will 'lock' if submission of the revision is attempted 21 or more days after the deadline. If you do not think you will be able to meet this deadline please contact the editorial office immediately.

on behalf of Kevin Padian (Subject Editor)
openscience@royalsociety.org

Associate Editor Comments to Author:

Comments to the Author:

In most circumstances, the journal does not permit multiple rounds of revision, but given the efforts you've already made (and the satisfaction of one reviewer that the paper is ready for acceptance), we're prepared to offer a final round of revision to you on condition that you do engage with the concerns raised by more critical referee. They have offered a couple of ways through for you, and it is really up to you how you respond, but a response is certainly needed - while the journal encourages authors to conduct replication-type studies, they need to have appropriate analytical and statistical treatment. We hope the reviewer's comments are helpful in guiding the final revision.

Reviewer comments to Author:

Reviewer: 1

Comments to the Author(s)

The revisions made have addressed my previous concerns.

Reviewer: 2

Comments to the Author(s)

I thank the authors for the responses and revisions, and provide some further comments below.

I think the analyses of genotype median and maximum are appropriate, since they are based on genotype-means (but this should be clarified in the methods).

It's unclear as to whether the authors use the individual fly or the genotype means as the unit of statistical analysis in some of their analyses (E.g. analyses on sex differences in longevity).

The new results of the proportional hazards model (cox model) indicate a very strong mating-by-sex interaction on survival; yet this was overlooked and not scrutinised further and not mentioned again in the Results sections on Longevity Cost of Reproduction (page 32, line 40). But, I have some comments on this Cox model and the general linear models, since these do seem to assume that every individual fly sampled was an independent data point, for which the reader will require clarification. I explain more below.

The authors argue against incorporating variance components that are currently unaccounted for in their cox and general linear models (such as Vial ID); arguing they could do this but that it would limit comparisons to other studies in the aging research community who never take account of such variance, and use similar statistical methods to those employed in this paper. I empathize with the authors on this point. But, models that account for such sources of variance are well suited to these analyses, and at least routinely used by researchers who study the evolutionary basis of ageing; and that vial-sharing effects are not just potential sources of variance, but tend to always account for a large part of the variance. Including them is required to make sure the data are not modelled at the incorrect level, resulting in pseudoreplication and anticonservative parameter estimates. The authors argument for why they don't take account of these random effects is interesting given a large premise of the current paper is to point out that previously assumed effects of cost of mating on longevity may have likely been driven small stochastic environmental changes leading to non-reproducibility of results across studies. Using

an approach that would accurately model and account for unaccounted for sources of variance would help very much in course correcting the field.

Notwithstanding, prior to these powerful approaches to multi-level modelling becoming available, researchers used to calculate vial means to conduct their analysis on these means rather than on data from individual flies (thus the unit of replication was vial rather than the fly, and the analyses not pseudoreplicated). This can't be done with survival analyses (they really require random effects), but could be done with linear models; and the analysis of genotype means does seem to be the approach used by the authors in the correlation analyses (but this should be made very transparent to the reader).

In sum, at the very least, the authors should be completely transparent and specify their units of replication for their analyses in their methods for all of their analyses – i.e. are they assuming all flies within each genotype were assayed as part of the same population (i.e. vial effects and other levels of the data in which flies within a genotype that by definition share inter-dependencies are assumed not to have an effect on survival), and should also be clear about the level of replication per genotype assumed in their analyses (seems that it is the level of the fly -- individuals per genotype in the cox models and linear models).

===PREPARING YOUR MANUSCRIPT===

===PREPARING YOUR REVISION IN SCHOLARONE===

To revise your manuscript, log into <https://mc.manuscriptcentral.com/rsos> and enter your Author Centre - this may be accessed by clicking on "Author" in the dark toolbar at the top of the

page (just below the journal name). You will find your manuscript listed under "Manuscripts with Decisions". Under "Actions", click on "Create a Revision".

Author's Response to Decision Letter for (RSOS-210273.R1)

See Appendix B.

RSOS-210273.R2 (Revision)

Review form: Reviewer 2

Is the manuscript scientifically sound in its present form?

Yes

Are the interpretations and conclusions justified by the results?

Yes

Is the language acceptable?

Yes

Do you have any ethical concerns with this paper?

No

Have you any concerns about statistical analyses in this paper?

Yes

Recommendation?

Accept with minor revision (please list in comments)

Comments to the Author(s)

I thank the authors for carefully considering my comments on the previous version, and their inclusions of analyses that control for the vial effect (coxme), and inclusion of discussion of this. Below are some final comments on the presentation of the Results, which the authors might like to take into account to broaden the accuracy, impact and scope of their work.

Results section – page 10, line 20

The p value doesn't indicate the strength of the effect. The authors should use the hazard ratio / odds ratio of the cox model when discussing the magnitude of the significant mating effects.

The same issue applies to reporting of correlations – if you indicate a strong correlation, you need to report the correlation coefficient (which indicates the strength) as well as the p value.

I don't think the sex x mating interaction is interpreted – we don't know if it is male or females survival that is more sensitive to mating. I am guessing females are more sensitive, but this needs to be clarified.

Clarify – the mating x sex x genotype interaction (and the respective 2 way interactions) were not statistically significant in the full cox model and were therefore removed?

Decision letter (RSOS-210273.R2)

Dear Dr Hoffman

On behalf of the Editors, we are pleased to inform you that your Manuscript RSOS-210273.R2 "Sex, mating, and repeatability of *Drosophila melanogaster* longevity" has been accepted for publication in Royal Society Open Science subject to minor revision in accordance with the referees' reports. Please find the referees' comments along with any feedback from the Editors below my signature.

Please submit your revised manuscript and required files (see below) no later than 7 days from today's (ie 20-Jul-2021) date. Note: the ScholarOne system will 'lock' if submission of the revision is attempted 7 or more days after the deadline. If you do not think you will be able to meet this deadline please contact the editorial office immediately.

on behalf of Prof Kevin Padian (Subject Editor)
openscience@royalsociety.org

Reviewer comments to Author:

Reviewer: 2

Comments to the Author(s)

I thank the authors for carefully considering my comments on the previous version, and their inclusions of analyses that control for the vial effect (coxme), and inclusion of discussion of this. Below are some final comments on the presentation of the Results, which the authors might like to take into account to broaden the accuracy, impact and scope of their work.

Results section – page 10, line 20

The p value doesn't indicate the strength of the effect. The authors should use the hazard ratio / odds ratio of the cox model when discussing the magnitude of the significant mating effects.

The same issue applies to reporting of correlations – if you indicate a strong correlation, you need to report the correlation coefficient (which indicates the strength) as well as the p value.

I don't think the sex x mating interaction is interpreted – we don't know if it is male or females survival that is more sensitive to mating. I am guessing females are more sensitive, but this needs to be clarified.

Clarify – the mating x sex x genotype interaction (and the respective 2 way interactions) were not statistically significant in the full cox model and were therefore removed?

===PREPARING YOUR MANUSCRIPT===

===PREPARING YOUR REVISION IN SCHOLARONE===

Author's Response to Decision Letter for (RSOS-210273.R2)

See Appendix C.

Decision letter (RSOS-210273.R3)

Dear Dr Hoffman,

I am pleased to inform you that your manuscript entitled "Sex, mating, and repeatability of *Drosophila melanogaster* longevity" is now accepted for publication in Royal Society Open Science.

on behalf of Kevin Padian (Subject Editor)
openscience@royalsociety.org

Appendix A

We thank the two anonymous reviewers for their insightful comments that have greatly improved our manuscript. We respond to each comment individually below in bold.

Reviewer comments to Author:

Reviewer: 1

Comments to the Author(s)

This manuscript would be improved by a more detailed explanation of the statistical methods written in a way a non-specialist can grasp what is being tested and how. This paper is likely to be read by many researchers using *Drosophila* as a model organism that do not routinely use these tests. Citations to these statistical methods should be included.

Also, I could not find any references to a "paired Wilcoxon test" when I was searching pubmed and the general internet - only to a paired Wilcoxon test. Was "Wilcoxon" test the correct name?

You are correct that this should have been written as a paired Wilcoxon test. It is written as `wilcox.test()` in R and as such we incorrectly wrote it in the manuscript

Figures 1, 2, and 4 should include units (days?) for longevity on the axes.

Thank you for catching this mistake. We have edited the figures accordingly.

Figure 3 should also include units on the axes.

We have updated all figures to make sure the axes are appropriate

For all figures - as there appears to be space, the spacing for each unit on the X and Y axis should be the same, producing a square rather than a rectangle, providing a clearer (more intuitive) comparison between conditions.

Fixed

Figures S1 and S2 should be included as full figures to support the text.

We debated from the beginning whether or not to include these figures in the main text, and ended up with them in the supplement due to their size. However, based on this comment, we have moved them back into the main text.

Reviewer: 2

Comments to the Author(s)

This is a very interesting manuscript, in which the authors analyse patterns of survival and longevity for virgin and mated flies, of each sex, across flies (*Drosophila melanogaster*) of 15 different genotypes. They do this across 2 cohorts (separated by 2 years in time, in which each cohort was measured in a different season and using a different diet substrate). The authors show that the genotypic effect on lifespan is strong across the two cohorts (strong correlation across genotypes, over Cohorts 1 and 2, and these measures are correlated to previous measures of genotype-specific longevity measured in other labs previously (by other researchers). What was surprising, however, is that there was no general effect of mating on longevity (the authors had predicted to observe a cost of reproduction, as gauged by reduced longevity of flies that had previously mated relative to virgin flies), and no general sex differences on longevity (the authors had predicted to observe a female-bias, typical of other studies, but if anything noted that a slight tendency of a male bias). The authors conclude that the effects of environmental variation (cohort-effects) on longevity render the study of sex differences and mating effects unreliable (subject to low levels of repeatability across time and across labs), hence promoting a cautionary tale for researchers examining the factors that

moderate patterns of aging in model species.

I read the paper with interest, but do have some comments that require careful attention.

Firstly, I thought that the statistical analyses may have failed to account for the hierarchical structure of the data (which would result in pseudoreplication in the analysis). Also, I am not sure why the authors can't model all factors (genotype, cohort, sex, mating status) in the one-and-the-same models (a proportional hazards model for survival, and then linear model for longevity). They should add random effects of vial identity (since ~ 20 individuals shared the same vial throughout the experiment, and these vial-sharing effects are always large and need accounting for, otherwise will in themselves lead to spurious results and interpretations of the results, since the models will treat each individual as an independent observation when they are not independent – i.e. this would lead to pseudoreplication) and ideally the mating group identity (flies were mated in groups of 100 prior to being allocated to a sex-specific vial). These analyses can be conducted in the packages `coxme` and `lme4` respectively.

We agree that the effects of vial number could *potentially* have had an effect on the results. However, this type of analysis -- using individual vial or plate or cage as a random variable -- is not commonly done in aging research, and as such, would prevent our work from being meaningfully compared with other studies such as those of the Kapahi, McCay, Partridge, Pletcher, and Tatar labs, all major players in aging research, and whose work we compared our own in this paper. For example, one study we compared our results to (Wilson et al 2020) used only a log rank test to compare within a genotype, across 8 replicate vials. In addition, the sample size required in that sort of analysis would not be feasible with normal resources. We are well-aware that there are microenvironments in the incubator/order of transfer that would lead to potentially vial to vial variation, which is why we randomize across the experiment. While we want this variation spread across all treatments, we do not want this variation to take up a large portion of our variance in our Cox models.

We could have modeled all the survival data in one large model; however, it would have made the direct comparisons across cohorts within genotypes much more difficult, as one genotype would set as a baseline, and we would more be asking is there an overall effect of genotype and a cohort by genotype interaction, which would allow us to make general observations about the data but not specific instances where the results were reproduced. To make it clear though, we did add in this large model to the very beginning of the results section showing that there was an effect of all variables analyzed. Similarly, for median lifespans, we added in a linear model looking at the effect of sex, genotype, mating, and cohort.

For correlations, it is unclear why the authors used the nonparametric Spearman Rank correlations and paired Wilcoxon tests rather than the parametric alternatives; particularly given that in the figures, the relationships look basically linear. Why was the alpha criterion set to 0.01 for Spearman rank tests – is this based on the number of correlations you tested (Bonferroni adjusted)?

We have clarified in the methods that a couple of the datasets had minor skews, and as such needed a non-parametric analysis. To keep things consistent and conservative, we used these non-parametric tests throughout. However, if we had used parametric analyses, the results would not be substantially different.

How did the authors handle escapees – flies that were lost to the experiment before they died (censored)?

We have added into the methods that flies that were censored were removed from the analysis, as this was very few flies in either cohort.

I think it is interesting that there was no clearly detectable cost of reproduction on longevity. Bearing in mind that the period of sexual cohabitation was relatively short (24 hours) and early in life (Day 2-3 of adult age, when median longevity is of the order of 40 to 60 days; what is missing here is a discussion of a) how long it takes females to become sperm limited (and discussion of whether this means reproductive costs are limited to a relatively short period early in life in which females are able to lay fertilizable eggs) and b) a discussion of the natural mating rate in fruit flies in the wild.

This is an interesting point that we thank the reviewer for bringing it up. We have added into the discussion a couple sentences about sperm limitation in females, as well as the natural rate of mating in the wild.

Discussion – in the first paragraph of the discussion, the authors talk about stability of effects across dietary regimes. This statement should be modified because there were several differences between the 2 cohorts (which the authors point out later in the discussion), so it is not possible to determine whether the cohort effects are primarily mediated by dietary variation or something else.

Thank you for pointing this out. We have reworded this to make it clear we were talking about stability of longevity from one cohort to the next and not the dietary regime.

The sentence at the top of page 11 requires rephrasing (particularly the phrase: “results failed to stay significant”).

We have rephrased this sentence to read “results were not consistent from one cohort to the next”

Appendix B

We thank the reviewer for their continued detailed comments. Below we address each point in bold.

It's unclear as to whether the authors use the individual fly or the genotype means as the unit of statistical analysis in some of their analyses (E.g. analyses on sex differences in longevity).

We have made it clear that for our Cox proportional hazard models flies were run as individuals, but for the Wilcoxon and correlation analyses across genotypes we used either the mean or median.

The new results of the proportional hazards model (Cox model) indicate a very strong mating-by-sex interaction on survival; yet this was overlooked and not scrutinised further and not mentioned again in the Results sections on Longevity Cost of Reproduction (page 32, line 40). But, I have some comments on this Cox model and the general linear models, since these do seem to assume that every individual fly sampled was an independent data point, for which the reader will require clarification. I explain more below.

We have updated the results to further indicate that there was a sex-by-mating interaction in the overall combined model which is present in some individual genotypes but not others. This was not surprising that an interaction effect was found in the entire model as the sample size at this point is so large, it will pick up any small difference in survival rate between the sexes of different mating statuses.

The authors argue against incorporating variance components that are currently unaccounted for in their Cox and general linear models (such as Vial ID); arguing they could do this but that it would limit comparisons to other studies in the aging research community who never take account of such variance, and use similar statistical methods to those employed in this paper. I empathize with the authors on this point. But, models that account for such sources of variance are well suited to these analyses, and at least routinely used by researchers who study the evolutionary basis of ageing; and that vial-sharing effects are not just potential sources of variance, but tend to always account for a large part of the variance. Including them is required to make sure the data are not modelled at the incorrect level, resulting in pseudoreplication and anticonservative parameter estimates. The authors' argument for why they don't take account of these random effects is interesting given a large premise of the current paper is to point out that previously assumed effects of cost of mating on longevity may have likely been driven small stochastic environmental changes leading to non-reproducibility of results across studies. Using an approach that would accurately model and account for unaccounted for sources of variance would help very much in course correcting the field.

Notwithstanding, prior to these powerful approaches to multi-level modelling becoming available, researchers used to calculate vial means to conduct their analysis on these means rather than on data from individual flies (thus the unit of replication was vial rather than the fly, and the analyses not pseudoreplicated). This can't be done with survival analyses (they really require random effects), but could be done with linear models; and the analysis of genotype means does seem to be the approach used by the authors in the correlation analyses (but this should be made very transparent to the reader).

In sum, at the very least, the authors should be completely transparent and specify their units of replication for their analyses in their methods for all of their analyses – i.e. are they assuming all flies within each genotype were assayed as part of the same population (i.e. vial effects and other levels of the data in which flies within a genotype that by definition share inter-dependencies are assumed not to have an effect on survival), and should also be clear about

the level of replication per genotype assumed in their analyses (seems that it is the level of the fly -- individuals per genotype in the cox models and linear models).

We have decided to do as the reviewer suggests and include the analysis both with and without controlling for the effects of vial. The results have been updated and the new Cox model results were added to the supplement (Table S2). Interestingly, controlling for vial reduces the significance of the mating effects much more than the sex effects. We have added a paragraph into the discussion that reflects the differences that can occur when controlling for between vial variation.

Appendix C

We again greatly thank the time reviewer 2 has put into our manuscript, and we think the paper is much better from their comments. We have addressed the minor comments below in bold.

Reviewer comments to Author:

Reviewer: 2

Comments to the Author(s)

I thank the authors for carefully considering my comments on the previous version, and their inclusions of analyses that control for the vial effect (coxme), and inclusion of discussion of this. Below are some final comments on the presentation of the Results, which the authors might like to take into account to broaden the accuracy, impact and scope of their work.

Results section – page 10, line 20

The p value doesn't indicate the strength of the effect. The authors should use the hazard ratio / odds ratio of the cox model when discussing the magnitude of the significant mating effects.

Thanks for pointing this out, we have added in the model coefficient.

The same issue applies to reporting of correlations – if you indicate a strong correlation, you need to report the correlation coefficient (which indicates the strength) as well as the p value.

Again, thank you for pointing this out. We have added in the rho values for the correlations in the first part, similar to with the comparison with previously published longevities.

I don't think the sex x mating interaction is interpreted – we don't know if it is male or females survival that is more sensitive to mating. I am guessing females are more sensitive, but this needs to be clarified.

It was actually the virgin males that had decreased survival to mated males overall. However, the effect size of the combined interaction is minor and genetic variation plays a considerably larger role. We have expanded on this in the results and discussion.

Clarify – the mating x sex x genotype interaction (and the respective 2 way interactions) were not statistically significant in the full cox model and were therefore removed?

We did not include the three-way interaction in our original analysis, as we were interested in the sex*mating interaction specifically. If we use the three way interaction, we do find a significant three way interaction, perhaps not unsurprisingly because of the large genetic variation in response as described in the single genotype analyses. Therefore, including it seemed to add little. We have kept the analysis as is for this reason.